

# Long-term evolution of preferences for conservation projects in the Seto Inland Sea, Japan: a comprehensive analytic framework

Takuro Uehara[1], Takahiro Tsuge[2] and Takahiro Ota[3]

[1] College of Policy Science, Ritsumeikan University, Ibaraki, Osaka, Japan
[2] Department of Economics, Konan University, Kobe City, Hyogo, Japan
[3] Graduate School of Fisheries and Environmental Science, Nagasaki University, Nagasaki City, Nagasaki, Japan

## ABSTRACT

**Background:** The long-term evolution of preferences for nature is crucial to conservation projects, given their targeted long-term horizons. Neglecting to account for this evolution could lead to undesirable human–nature relationships. This study compares the willingness to pay (WTP) for three coastal conservation projects in the Seto Inland Sea, Japan, at two distant time points (1998 and 2015), and tests for temporal transferability. It also compares protest responses that are often overlooked in WTP practices, regardless of their utility for conservation projects.
**Methods:** Given the lack of a unanimous protocol for protest response analyses and their use in estimating WTP, we propose a comprehensive analytic framework that integrates the two.
**Results:** We show that, while preferences for coastal ecosystem services were overall stable and temporarily transferable, the preferences for certain aspects of conservation projects considerably changed.
**Discussion:** This suggests the need to reconsider the projects' scheme, not the ecosystem services themselves, along with the clarification of beneficiaries and those responsible for past destruction. We conclude by suggesting further studies with a focus on regions experiencing significant social-ecological changes, such as developing countries, by exploiting the rich asset of existing valuations. This could contribute to the database for more temporal-sensitive ecosystem service valuations utilized for benefit transfers.

# INTRODUCTION

Understanding people's valuations of nature and how they change in the long run is of crucial importance to establishing and sustaining the desired relationships with nature (*Uehara et al., 2016*; *Uehara & Mineo, 2017*). The potential evolution of preferences for nature indicates that a conservation project assuming constant preferences could lead

Corresponding author
Takuro Uehara,
takuro@fc.ritsumei.ac.jp

us to an undesired state. Ideally, conservation projects should be adaptive to evolution (*Skourtos, Kontogianni & Harrison, 2010*).

A key approach to understanding how people value nature (i.e., welfare) is measuring the willingness to pay (WTP) for ecosystem services, whose techniques have been well-developed in environmental economics (*Gómez-Baggethun et al., 2010*; *Freeman, Herriges & Kling, 2014*). However, there is scope for improvement in WTP-based studies; in particular, two aspects could foster better contributions to the literature. First, WTP studies generally conduct one-time estimates, relative to the time horizon for conservation projects, the related research on the evolution of WTP is short termed (e.g., from few weeks to a year) (*Skourtos, Kontogianni & Harrison, 2010*). These short-term studies tend to focus on stability and equality rather than evolution or changes (*Jakus, Stephens & Fly, 2005*). Moreover, they often assume the temporal stability of WTP, rather than conducting explicit tests (*Brouwer & Bateman, 2005*; *Costanza et al., 1997*, *2014*) or using simple variations in previous WTP estimates for future projections (*Kubiszewski et al., 2017*). Second, WTP estimates commonly exclude protest respondents (*Brouwer & Martín-Ortega, 2012*), that is, respondents who reject certain aspects of a conservation project presented in a survey by saying "no" to a proposed bid for the project, even though they positively value the ecosystem services (*Freeman, Herriges & Kling, 2014*). Protest responses provide non-negligible information for conservation projects in a real-world context, not in a vacuum (*García-Llorente, Martín-López & Montes, 2011*). While WTP reveals preferences for ecosystem services that benefit from conservation projects, protest response analyses highlight preferences for project design and implementation. Reviewing past environmental valuation studies, *Meyerhoff & Liebe (2010)* found that, on average, the rate of protest responses is 17.69%, indicating a simple disposal could result in a significant loss of information. In addition, it could lead to a biased WTP estimate if people who protest systematically differ from those who do not (*Brouwer & Martín-Ortega, 2012*; *Freeman, Herriges & Kling, 2014*).

Our study aims to understand the long-term evolution of preferences for coastal ecosystem services by addressing the abovementioned, underdeveloped yet crucial topics: evolution of welfare measured in WTP and that of protest responses. Since common WTP practices exclude protest responses and there is no unanimous protocol on how to deal with them (*Meyerhoff & Liebe, 2010*), we propose a comprehensive analytic framework that integrates WTP estimation and protest response analyses and comprises five research questions. We compare the coastal and non-coastal residents' preferences for three hypothetical projects that provide coastal ecosystem services in the Seto Inland Sea (SIS), Japan, at two distant time points, 1998 and 2015. A 17-year difference is sufficient to include next generations that were not included in the 1998 survey.

## MATERIALS AND METHODS

Figure 1 presents a comprehensive analytic framework with the five research questions. While common practices focus on the temporal comparison of WTP at two time points (RQ 4) as well as the temporal transferability of WTP and functions of WTP estimates

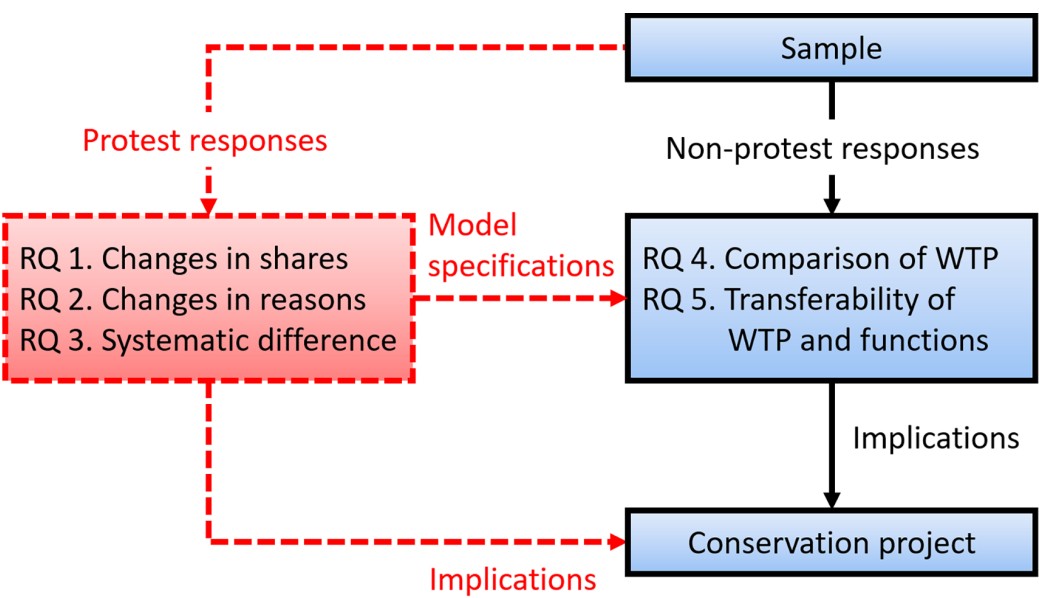

**Figure 1 Comprehensive analytic framework for the evolution of preferences for conservation projects.** The red dashed lines indicate a procedure overlooked in common WTP practices and benefit transfers.

(RQ 5) (*Downing & Ozuna, 1996*; *Brouwer & Spaninks, 1999*; *Brouwer & Bateman, 2005*; *Zandersen, Termansen & Jensen, 2007*; *Rosenberger, 2015*), the present framework adds three research questions: how have the shares of protest responses changed (RQ 1); how have the reasons for protest responses changed (RQ 2); and is there a systematic difference between protesters and non-protesters (RQ 3)? RQ 3 could provide important information about model specifications that could elicit unbiased WTP, as discussed later.

## Three hypothetical projects in the Seto Inland Sea

The three hypothetical projects were designed to elicit WTP for coastal ecosystem services in the SIS, an enclosed coastal sea in western Japan (Fig. 2). The SIS was rich in ecosystem services; however, these ecosystems were destroyed or degraded with the rapid economic progress since the mid-20th century, resulting in, for example, declining fish catches, destruction of coastal zones for landfills and other anthropocentric uses, and water pollution (*The Association for the Environmental Conservation of the Seto Inland Sea, 2015*).

The projects include the restoration of the natural beauty of coastlines (Project 1), conservation of seagrass beds as cradles of the sea (Project 2), and protection of natural coastlines through a national trust (Project 3) (see Supplementary Information S1 for more details). We consider the same three projects in both 1998 and 2015; however, owing to certain changes in the SIS, we present changes in the hypothetical projects in the 2015 survey.

## Data generating processes

An internet survey was conducted in 1998 and 2015, in which coastal and non-coastal residents were asked to respond to a questionnaire on WTP estimates for the three
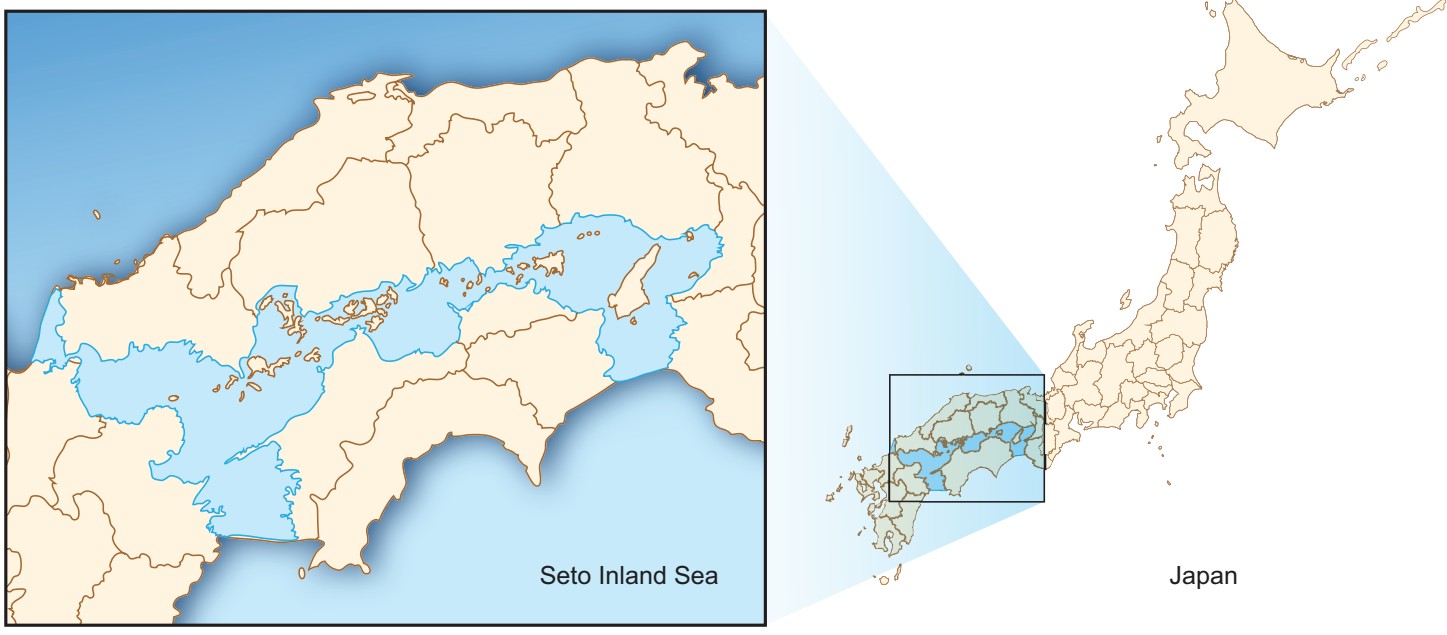

Figure 2 **Location of Seto Inland Sea, Japan.**

projects. For the 1998 survey, we utilized raw data collected by *Tsuge & Washida (2003)*. The survey website was posted at the top page of a national newspaper website, Asahi Shimbun (www.asahi.com). An average of 440,000 people visited the website per month. The survey was conducted between December 1 and 12, 1998. The 2015 survey was also posted at Asahi Shimbun, and we requested an online internet survey company to collect the sample to ensure the sample was sufficiently comparable with the 1998 survey. The survey was conducted between December 2 and 7, 2015.

## Protest response analysis

For the protest response analysis, we used the statistical analysis software STATA (Version 14.2) by StataCorp LP (http://www.stata.com).

### Coding protest responses

Information on protest responses was generated from the reasons for rejecting bids for projects in the questionnaire. Respondents could either choose from the reasons available or provide an independent answer. To conduct quantitative analysis, we coded the open answers and created eight categories, as explained in the "Results" section. Some reasons for rejection were valid and, thus, not considered a protest response (e.g., "I support these projects, but the contribution amount is too high."). Since there is no clear-cut definition of protest responses (*Brouwer & Martín-Ortega, 2012*), we follow discussions in well-established textbooks on the valuation of the environment (*Freeman, Herriges & Kling, 2014*; *Rosenberger & Loomis, 2017*) to choose protest responses from reasons coded in the survey.

### Detection of systematic difference

While we note the potential impact of systematic differences between protestors and non-protestors on WTP estimates (*Freeman, Herriges & Kling, 2014*), there is no unanimous protocol to address such differences (*Meyerhoff & Liebe, 2010*). Here, we chose a logit model for a binary response to detect the systematic difference. That is,

$$\Pr(y_i = 1 | \boldsymbol{x}_i) = \frac{e^{\boldsymbol{x}_i \boldsymbol{\beta}}}{1 + e^{\boldsymbol{x}_i \boldsymbol{\beta}}}, \tag{1}$$

where $y_i$ is a binary response regarding whether respondent $i$ is a protestor ("1") or non-protestor ("0"). $\boldsymbol{x}_i$ is a vector of explanatory variables and $\boldsymbol{\beta}$ is a vector of coefficients. With the logit model, we can identify factors that determine whether a respondent is a protestor.

## Welfare analysis

To analyze dichotomous choice-contingent valuation data (Version 0.0.15), we used a package by *Nakatani, Aizaki & Sato (2016)* run on R (Version 3.3.2 for Windows (64 bit)) (*R Core Team, 2016*).

### WTP estimate

We used a single-bounded dichotomous choice format, which was used in the 1998 study (*Tsuge & Washida, 2003*). It is less susceptible to bias than open-ended or payment card formats (*Mitchell & Carson, 1989*). As bids, each respondent was shown one among six randomly selected amounts: 500, 1,000, 3,000, 8,000, 15,000, and 30,000 JPY. The respondents were then asked if they were willing to pay the amount toward the implementation of each project. We assumed the payment would be made only once. To ensure that the respondents recognized the payment burden, we explained that the donation amount would be deducted from the money used for other household purposes. Those who agreed to donate the amount were asked to specify the expenditures they forfeit for the donation.

The response data were analyzed using the binary logit model derived from the random utility model (*Hanemann, 1984*). In the model, the following is assumed as the utility respondent $k$ obtains from the alternative:

$$U_{ki} = V_{ki} + \varepsilon_{ki}, \tag{2}$$

where $i$ takes the symbol $y$ when respondent $k$ answers "yes" to the bid and $n$ when respondent $k$ answers "no." $V_{ki}$ and $\varepsilon_{ki}$ represent the observable deterministic term and unobservable error term of utility. It is assumed that respondent $k$ considers cost and environmental improvement realized by the conservation project and chooses an alternative with higher utility. The probability $P_{ky}$ that the respondent $k$ will answer yes is equal to the probability that the utility from the alternative $U_{ky}$, is larger than the utility from the alternatives $n$, $U_{kn}$, as described below:

$$P_{ky} = Pr(U_{ky} > U_{kn}) = Pr(V_{ky} + \varepsilon_{ky} > V_{kn} + \varepsilon_{kn}). \tag{3}$$

Assuming error term $\varepsilon_{ki}$ follows a type-I extreme value distribution (Gumbel distribution), probability $P_{ky}$ is described by the following binary logit model:

$$P_{ky} = \frac{1}{1 + e^{-\Delta V}}, \tag{4}$$

where $\Delta V$ denotes the utility difference function and the following linear function is assumed: $\Delta V = \alpha + \beta T_k$. In the utility difference function, $T_k$ represents the bid offered to respondent $k$ and $\alpha$ and $\beta$ indicate the utility obtained from environmental improvement and utility obtained from the payment. By extending the utility difference function as follows, it is possible to analyze the influence of other factors (e.g., household income) on the respondents' answers: $\Delta V = \alpha + \beta T_k + \gamma z_k$, where, $z_k$ is a vector of other factors, possibly affecting respondent $k$'s answer and $\gamma$ is a vector of parameters for those factors.

The parameters are estimated by the maximum likelihood method (*Greene, 2012*). The log likelihood function can be written as follows:

$$lnL = \sum_k \sum_i \delta_{ki} lnP_{ki}, \tag{5}$$

where $\delta_{ki}$ is a dummy variable, such that $\delta_{ki} = 1$ when respondent $k$ answers "yes" to a bid, and $\delta_{ki} = 0$ otherwise.

The mean WTP can be calculated using the estimated parameters, $\alpha$ and $\beta$ (*Hanemann, 1984*). It is obtained by integrating the probability that the respondent will answer "yes" to the bid. However, since it is not realistic to integrate an extremely high amount, the maximum bid is often used as the integration upper limit. In this case, the mean WTP is calculated as follows:

$$\text{Mean WTP(truncated at } T_{\max}) = \int_0^{T_{\max}} P_{ky} dT, \tag{6}$$

where $T_{\max}$ is the maximum bid.

### Confidence intervals

We calculated the confidence intervals using Krinsky and Robb's technique (*Krinsky & Robb, 1986*), which is often employed in stated preference methods, such as the contingent valuation method (CVM) and conjoint analysis (*Downing & Ozuna, 1996*; *Zandersen, Termansen & Jensen, 2007*; *Lew & Wallmo, 2017*; *Matthews, Scarpa & Marsh, 2017*). Using the technique, we draw 10,000 random coefficients and compute 10,000 mean WTP measures. Then, we ordered the 10,000 mean WTP measures from the smallest to largest and selected the 95% confidence limits.

### Transferability test

Since it is impossible for people who are not born yet to report their future WTP and confirm the extent to which a current project will be supported by future generations, we need to extrapolate a future WTP value by exploiting value information currently available. A method that has been widely used is benefit transfers, which involve transferring existing value information to a new context (*Rosenberger & Loomis, 2017*). There are two primary types of benefit transfers: value and function transfers.

Value transfers are the direct application of summary statistics in existing research such as per unit measure of WTP (*Rosenberger & Loomis, 2017*). It generally assumes constant preferences over time (*Costanza et al., 1997*, *2014*). Function transfers tailor value estimates by reflecting differences in the characteristics of contexts in a model estimating WTP.

In addition to a strong interest in the transferability of WTP estimates given the scarcity of resources, the temporal stability of the estimates and its testing methods have been extensively studied, although most studies are limited to the short term (from a week to two years) (*Skourtos, Kontogianni & Harrison, 2010*). The key focus is the statistical equality of WTP and coefficient parameters of models using various statistical tests such as the *t*-test, Wald test, likelihood ratio test, Mann–Whitney test, and Kolgorov–Smirnov test (*Brouwer & Spaninks, 1999*).

However, we did not conduct these statistical tests for two reasons. First, the statistical tests examine statistical equality and ignore acceptable levels of accuracy in a real-world context (*Rosenberger, 2015*). A review by *Rosenberger (2015)* reveals that most studies failed to pass these tests. Second, the coefficients estimated by the logit model used in this study are not purely parameters of the utility function but products of parameters of the utility function and scale parameter (*Train, 2009*). Therefore, testing the statistical equality of the estimated coefficients does not necessarily mean examining the statistical equality of the parameters of the utility function and there is a possibility of erroneous judgments on the latter. On the other hand, since WTP estimates are calculated from the ratio of estimated coefficients, the scale parameters of the numerator and denominator are canceled out and not affected by them. Therefore, it is more meaningful to test for the statistical equality of WTP estimates.

Hence, we evaluated the performance of value and function transfers by conducting a percentage transfer error (PTE) test, which is a type of transfer error test (*Rosenberger, 2015*) that measures the difference between the benefit transfer value (estimated using 1998 values or functions) and true value (2015 estimates). While the abovementioned tests focus on equality, this test estimates maximum transfer error. The percentage transfer test is calculated as

$$\text{PTE} = \left[ \frac{V_T - V_P}{V_P} \right] \times 100, \tag{7}$$

where $V_T$ is the transfer estimate and $V_P$ the known or actual estimate for the policy site. PTE then measures the degree of difference between the transferred and actual estimates at the policy site. Typically, PTE requires both estimates to be available within the context of a primary study that has derived them (*Rosenberger, 2015*, p. 309).

For a function transfer, we used models that include income as an explanatory variable and incorporate average income for 2015 in the 1998 models for estimation. There are three reasons to include income as an explanatory variable. First, it is consistent with the economic theory underpinning this method (*Brouwer & Bateman, 2005*; *Hanemann, 1984*). Second, it is statistically significant (*Brouwer & Bateman, 2005*) in many empirical studies, including the present analysis. Finally, long-term income projection has been

**Table 1 Shares of protest responses by year and geographical origin.**

|  | Project 1 | | Project 2 | | Project 3 | |
|---|---|---|---|---|---|---|
|  | 1998 | 2015 | 1998 | 2015 | 1998 | 2015 |
| Coastal residents | 24.4% | 37.3% | 20.4% | 32.6% | 17.5% | 33.7% |
| Non-coastal residents | 25.8% | 41.6% | 22.4% | 39.0% | 18.9% | 38.2% |
| Total | 25.7% | 41.3% | 22.2% | 38.6% | 18.8% | 37.9% |

well-studied and available from various sources such as government agencies and the Organization for Economic Co-operation and Development.

## RESULTS

While the sample sizes have the same order of magnitude (5,632 respondents for 1998 and 7,264 respondents for 2015), there are significant differences in the rates of internet accessibility, which could affect the compositions of the samples (Supplementary Information S2 for descriptive statistics) and the following analyses. In Japan, personal accessibility to the Internet significantly increased from 13.4% in 1998 to 83.0% in 2015 (*Ministry of Internal Affairs and Communications, 2017*). While company employees account for the largest share of respondents in both 1998 and 2015, which is consistent with population characteristics, there are certain differences between the years. For instance, university students with internet access accounted for 14% of the respondents in 1998 but only 6% in 2015. Part-time workers, the unemployed, and housewives with internet access accounted for 3% of the respondents in 1998 and 36% in 2015. We calculated confidence intervals using the Krinsky–Robb technique in the following analysis on welfare changes and temporal transferability.

### Protest responses
#### RQ 1: share of protest responses
The share of protest responses is greater in 2015 than in 1998 and larger for non-coastal residents compared with coastal residents for all three plans (Table 1). The analysis of variance shows that these differences in the shares by year and geographical origin are statistically significant at the 10% level for all projects.

#### RQ 2: reasons for protest responses
We coded the protest responses into six types on the basis of multiple choices and open answers to reasons underpinning the rejection of a bid proposed for the projects. Here, we show the categorization of reasons by year and geographical origin (Tables 2–4). The patterns are similar across all projects, except for the change in Reason 1 for non-coastal residents.

The share of coastal residents increased for all three projects and they accounted for the second highest number of respondents protesting contribution to a fund (Reason 1). The respondents were asked to contribute to a newly established local fund (The SIS Environment Conservation Fund) to implement the projects. In the open answers, some

**Table 2  Composition of reasons for protest by year and geographical origin for Project 1.**

| Reason to oppose bid | Coastal residents | | | Non-coastal residents | | |
|---|---|---|---|---|---|---|
| | 1998 | 2015 | *t*-test | 1998 | 2015 | *t*-test |
| 1. I support these projects, but I am against contributing to a fund. | 19% | 33% | *** | 23% | 27% | *** |
| 2. I support these projects, but I don't think I need to personally take responsibility for funding. | 42% | 40% | | 37% | 54% | *** |
| 3. I am opposed to the program itself. | 34% | 25% | ** | 33% | 18% | *** |
| 4. I do not trust the survey. | 3% | 1% | | 3% | 0% | *** |
| 5. Information is insufficient to make a judgment. | 1% | 2% | | 4% | 1% | *** |
| 6. Did not understand the questionnaire. | 1% | 0% | | 1% | 0% | *** |
| Total | 100% | 100% | | 100% | 100% | |
| *N* | 96 | 170 | | 1,330 | 2,762 | |

**Notes:**
The *t*-tests examine the null hypothesis of no difference in numbers of times each reason was cited between 1998 and 2015.
** $p < 0.05$.
*** $p < 0.01$.

**Table 3  Composition of reasons for protest by year and geographical origin for project 2.**

| Reason to oppose bid | Coastal residents | | | Non-coastal residents | | |
|---|---|---|---|---|---|---|
| | 1998 | 2015 | *t*-test | 1998 | 2015 | *t*-test |
| 1. I support these projects, but I am against contributing to a fund. | 20% | 38% | *** | 28% | 29% | |
| 2. I support these projects, but I don't think I need to personally take responsibility for funding. | 46% | 49% | | 41% | 60% | *** |
| 3. I am opposed to the program itself. | 26% | 11% | *** | 23% | 10% | *** |
| 4. I do not trust the survey. | 4% | 1% | ** | 4% | 0% | *** |
| 5. Information is insufficient to make a judgment. | 3% | 1% | | 4% | 1% | *** |
| 6. Did not understand the questionnaire. | 1% | 0% | * | 1% | 0% | *** |
| Total | 100% | 100% | | 100% | 100% | |
| *N* | 80 | 149 | | 1,155 | 2,595 | |

**Notes:**
* $p < 0.10$.
** $p < 0.05$.
*** $p < 0.01$.

respondents stated that it should be funded from tax revenues because it is a public good whose cost should be incurred by everyone.

In both years, coastal and non-coastal residents accounted for the highest numbers in terms of the belief that funding was not their personal responsibility (Reason 2). In particular, the number of non-coastal residents significantly increased for Reason 2 and accounted for a larger share of protest responses for 2015. In the open answers, both coastal and non-coastal residents claimed the project(s) should be funded by people who are responsible for the environmental destruction, such as private companies and

**Table 4 Composition of reasons for protest by year and geographical origin for Project 3.**

| Reason to oppose the bid | Coastal residents | | | Non-coastal residents | | |
|---|---|---|---|---|---|---|
| | 1998 | 2015 | *t*-test | 1998 | 2015 | *t*-test |
| 1. I support these projects, but I am against contributing to a fund. | 22% | 38% | *** | 30% | 29% | |
| 2. I support these projects, but I don't think I need to personally take responsibility for funding. | 51% | 47% | | 44% | 59% | *** |
| 3. I am opposed to the program itself. | 20% | 11% | ** | 18% | 10% | *** |
| 4. I do not trust the survey. | 3% | 2% | | 3% | 1% | *** |
| 5. Information is insufficient to make a judgment. | 3% | 2% | | 3% | 1% | *** |
| 6. Did not understand the questionnaire. | 1% | 0% | * | 1% | 0% | *** |
| Total | 100% | 100% | | 100% | 100% | |
| *N* | 69 | 154 | | 974 | 2,538 | |

Notes:
* $p < 0.10$.
** $p < 0.05$.
*** $p < 0.01$.

municipalities. A characteristic unique to non-coastal residents is that while they valued these projects, some preferred to conserve the environment closer to their place of residence.

The rate of respondents opposed to the program itself (Reason 3) was lower in 2015. In the open answers, certain respondents who chose Reason 3 stated they were dubious about the effectiveness of the project(s). For example, some pointed out that the scales of the projects are too small to realize the benefits mentioned.

### RQ 3: systematic difference between protestors and non-protestors

Before building a model to estimate WTP, we tested the systematic difference between protestors and non-protestors. The WTP estimate could be biased if there is a systematic difference (*Freeman, Herriges & Kling, 2014*). However, there is no unanimous protocol for the treatment of protest responses (*Tobarra-González, 2015*). Here, we adopted a logit model to explore factors influencing a respondent's choice to protest or not. We chose place of residence, income, and year as explanatory variables. Because of the 17-year gap between 1998 and 2015, the samples were considered to be drawn from different populations. Given that respondents are geographically located in different areas, differentiating WTP by place of residence could also be informative for conservation projects (e.g., more targeted fundraising). Income is a key variable in economic theory (*Hanemann, 1984*; *Brouwer & Bateman, 2005*). The results revealed (Table 5) that all three variables explain the respondents' choice to protest at the statistically significant levels, indicating the possibility of a systematic difference between protestors and non-protestors by place of residence, income, and year. Therefore, it would be desirable to estimate WTP by constructing models on the basis of these three variables. However, since income has 15 categories and it is not realistic to model each category separately, we use income as an explanatory variable. Accordingly, we constructed four models for each project, resulting in a total of 12 models.

**Table 5 Logit models for three projects.**

| | Project 1 | | Project 2 | | Project 3 | |
|---|---|---|---|---|---|---|
| | Coef. | *t*-stat | Coef. | *t*-stat | Coef. | *t*-stat |
| Coastal_dummy | −0.164 | −1.95* | −0.241 | −2.74*** | −0.173 | −1.93* |
| Income | −0.014 | −2.36** | −0.015 | −2.42** | −0.020 | −3.03*** |
| Year | 0.035 | 13.92*** | 0.041 | 15.53*** | 0.051 | 18.62*** |
| Constant | −71.479 | −14.01*** | −82.404 | −15.63*** | −102.785 | −18.73*** |
| N | 10,933 | | 10,938 | | 10,937 | |
| Log-likelihood | −6,808.4 | | −6,537.5 | | −6,244.7 | |

**Notes:**
* $p < 0.10$.
** $p < 0.05$.
*** $p < 0.01$.

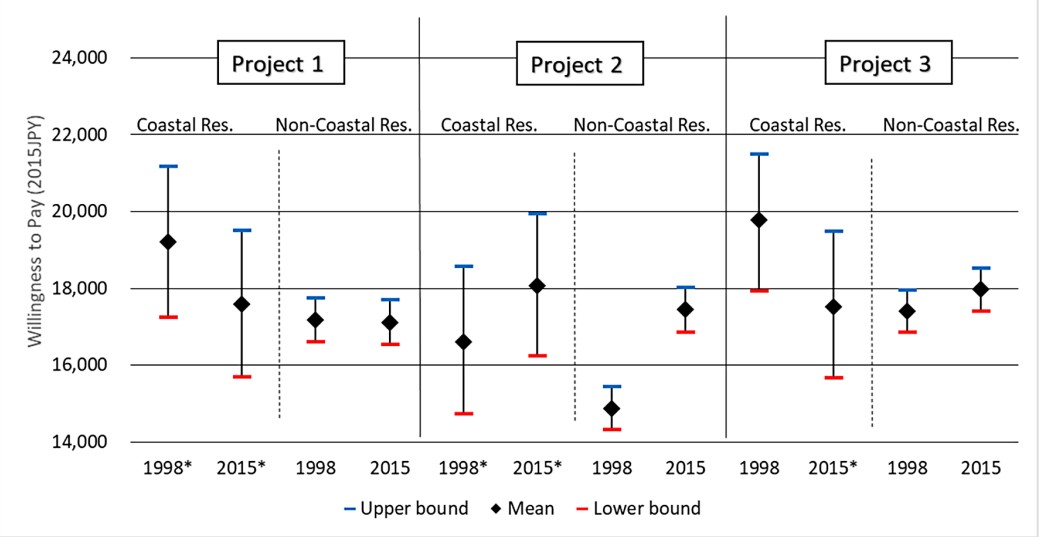

**Figure 3 Confidence intervals of mean WTP.** *Indicates income was excluded from the model because it was not statistically significant (Supplementary Informations S3 and S4).

## Welfare

### RQ 4: confidence intervals

This research question addresses the extent to which preferences for coastal ecosystem services (i.e., welfare obtained from the services) have evolved over the 17 years by measuring changes in people's WTP for the projects. On the basis of the protest response analysis, we built models to estimate WTP by year and geographical origin with income as an explanatory variable when it is statistically significant (Supplementary Information S3).

Figure 3 shows changes in the mean WTP with 95% confidence intervals across 17 years by geographical origin. The sample sizes differ significantly by geographical origins but have the same order of magnitude across time: 278–308 respondents for coastal residents and 3,146–3,772 respondents for non-coastal residents. The confidence intervals were relatively wider for coastal residents due to their smaller sample sizes. The confidence

**Table 6 Absolute percentage transfer errors with ranges.**

| | Residency | Value transfer | | Function transfer | |
|---|---|---|---|---|---|
| | | Mean \|PTE\| | Range | Mean \|PTE\| | Range |
| Project 1 | Coastal | 9.3 | 8.4–9.8 | **3.5** | **1.7–4.7** |
| | Non-coastal | **0.4** | **0.3–0.4** | 4.3 | 3.8–4.6 |
| Project 2 | Coastal | **8.2** | **6.8–9.2** | 12.0 | 8.9–15.4 |
| | Non-coastal | **14.8** | **14.3–15** | 18.0 | 17.1–18.5 |
| Project 3 | Coastal | 12.8 | 10.3–14.4 | **6.8** | **6.3–6.8** |
| | Non-coastal | **3.2** | **3–3.1** | 5.5 | 5.1–5.8 |

**Notes:**
Better transfers by project and residency are in bold. The range is from the minimum to maximum difference of WTP (mean, lower bound, and upper bound WTPs), between 1998 and 2015.

intervals among the same geographical origin are comparable because of the similar sample sizes.

The confidence intervals overlap for all models, except for non-coastal residents in the case of Project 2 (conservation of seagrass beds as cradles of the sea), indicating that only welfare obtained from Project 2 increased for non-coastal residents at the statistically significant level. The mean WTP increased by 17.4%, from 14,870 JPY in 1998 to 17,452 JPY in 2015. This change is also the largest among the point estimates of the mean WTP: −8.5% for coastal residents in Project 1, −0.4% for non-coastal residents in Project 1, 8.9% for coastal residents in Project 2, −11.3% for coastal residents in Project 3, and 3.3% for non-coastal residents in Project 3 (Supplementary Information S4). The non-coastal residents' mean WTP shows a stark contrast with their WTPs for the rest whose mean WTPs barely changed (−0.4% and 3.3%).

### RQ 5: temporal transferability

The research question is based on the extent of transferability of WTP estimates and models in 1998 to those in 2015. Table 6 presents the absolute PTEs for value and function transfers. PTE measures the difference in percentage between true values calculated using the 2015 data and model and transferred values are estimated using value information for 1998. Of the six transfers, four transfers performed better for the value transfer.

## DISCUSSION

### Protest responses

Protest response analyses are generally beyond the scope of WTP practices and benefit transfers. However, our study revealed that it provides non-negligible information on successful conservation projects that are implemented in the real world, not in a vacuum. Changes in the share and composition of protest responses demonstrated those in preferences for other aspects of a conservation project rather than the value of ecosystem services measured in WTP.

Overall, these shares are larger than the average share of protest responses in previous studies (mean: 17.69%; standard deviation: 11.30; median: 16.13; min.: 0; max.: 59.28)

(*Meyerhoff & Liebe, 2010*). The shares increase in 2015, which indicates the growing importance of protest response analyses as a source of information for conservation projects. In addition to the possibility of an actual increase in the protest responses in the 2015 population, the drastic changes in internet accessibility resulted in biased samples with varying population attributes. However, since there appears to be no study on temporal changes in the share of protest responses, we are still unaware if an increase in the share of protest responses is a general trend and of the factors influencing the increase. We leave this to further study.

The analysis of the reasons for protest highlights the need for policymakers to be adaptive and rethink the manner in which projects are implemented. More specifically, there is a growing dislike for payment methods (i.e., establishing a fund) (Reason 1). In addition, respondents who do not want to personally take responsibility for funding (Reason 2) account for the highest number of protestors, suggesting the reconsideration of the payment method along with a reflection of those responsible for the past destruction of ecosystems and project beneficiaries. Failure to account for these reasons could lead to policymakers facing unexpected oppositions at the time of actual project implementation, even if the WTP estimates that do not include protest responses indicate the projects as valuable. The choice of payment method is an important aspect of a project scheme (*Freeman, Herriges & Kling, 2014*) and this choice should be sensitive to the social context (*Fischhoff & Furby, 1988*). In addition to the method of payment, the recipient of these payments is an important aspect warranting consideration given that, according to protest responses, some non-coastal residents prefer spending money on a similar project closer to their place of residence (Reason 2).

The protest response analysis was also informative in correcting systematic differences caused by the exclusion of the protest responses from the WTP estimate. The logit models identified place of residence, income, and year as a source of systematic differences. While place of residence and year seem to be straightforward, further study is recommended to explain why income can be a significant explanatory factor for why respondents cast protest votes to draw larger implications. To obtain an accurate WTP estimate, we recommend the protest response analysis, especially when the share is not as small as that in our case study.

## Welfare

The confidence intervals showed that the welfare obtained from the projects measured in WTP was stable over the 17 years, except for non-coastal residents in the case of Project 2 (conservation of seagrass beds as cradles of the sea). Because studies of long-term evolution of WTP for a specific site are lacking, it is difficult to determine whether our finding is unique or conforms to other sites as well. However, these changes have the same order of magnitude as global estimates by *Costanza et al. (2014)*, who used the world database for valuation studies on ecosystem services: the unit value of estuaries decreased by 8.2% (from 31,509 USD/ha per year in 1997 to 28,916 USD/ha per year in 2011) and the unit value of seagrass or algae beds increased by 10.1% (from 26,226 USD/ha per year in 1997 to 28,916 2007 USD/ha per year in 2011). However, *Pendleton et al. (2016)*

conducted a closer examination of the data compiled by *Costanza et al. (2014)* and highlighted the lack of accuracy and comprehensiveness, especially for marine and coastal areas. For example, the database includes estimates from more than 20 years ago by assuming the temporal stability (or non-changing) of a unit value. This supports the importance of primary studies on the temporal valuation of marine and coastal ecosystem services.

The changes in WTP can be explained by factors affecting demand or supply of the ecosystem services (*Skourtos, Kontogianni & Harrison, 2010*). Factors affecting demand could include income, prices of other goods, and socioeconomic profile, while those influencing supply may be the amount and quality of ecosystem services.

There are three possible reasons for a higher WTP for Project 2 among non-coastal residents. First, while Project 2 was the same in both 2015 and 1998, the context was different. The 1998 survey presented the *possibility* of the seagrass bed being destroyed: "Moreover, 50 hectares of the largest remaining seagrass bed are currently proposed for reclamation as airport and harbor construction progresses." However, since the seagrass bed was destroyed by 2015, the corresponding survey addressed it as *actually* destroyed. This can be considered a scarcity signal. Previous studies also show the sensitivity of WTP to changes in the supply of ecosystem services in the SIS (*Tokiyoshi et al., 2005*). Second, the first reason may attract attention from those who do not live in coastal zones because they benefit from the airport and harbor construction. Finally, the restoration of the seagrass bed became a national agenda, and the Fisheries Agency launched an investigative committee for seagrass beds and mudflats around six months prior to our survey. There is no significant difference between Project 2 on one hand, and Project 1 and 3 on the other regarding the changes in protest responses probably because other factors, such as income and the various temporal factors shown in the logit models (Table 5) outweigh the three reasons mentioned above.

The PTEs were not large compared with those in previous studies: the mean of the mean PTE was 140 for the value transfer and 65 for the function transfer (*Rosenberger & Loomis, 2017*). The performance of a transfer is considered to depend on contextual similarity (*Rosenberger & Loomis, 2017*). Since these previous studies are about a spatial transfer (i.e., between spatially different sites) and not a temporal transfer (i.e., between temporally different but spatially same sites), the contextual difference resulting from the 17-year gap is smaller than the spatial differences in previous studies (*Rosenberger & Loomis, 2017*). It is difficult to judge whether these transfer errors are small among long-term temporal transfer studies since such studies are limited. *Zandersen, Termansen & Jensen (2007)* conducted a study on forest ecosystem services in Denmark and reported a PTE of 25 for 52 forests across a 20-year period. *Boman et al. (2011)* estimate this value at 17 for Sweden.

In contrast to general tendency (*Rosenberger, 2015*; *Rosenberger & Loomis, 2017*), in this study, the value transfer performs better than the function transfer: of the six transfers, four are better as value transfers. There are two possible explanations: temporal contextual similarity and insufficiency of the function transfer. First, as

*Bateman et al. (2011*, p. 383*)* argued, "the choice of (value vs. function transfer) depends crucially upon the degree of similarity of the sites under consideration." As the comparison of previous studies on spatial transfer revealed, study sites during 1998 and 2015 seemed similar. Second, our function transfer did not sufficiently capture changes because in general, function transfers perform better than value transfers as the former can increase transfer accuracy by reflecting site characteristics (*Rosenberger & Loomis, 2017*). In technical terms, there are two types of changes that affect WTP estimates. First are changes in the WTP estimate model's arguments and the second are those in the coefficients of the model (*Whitehead & Hoban, 1999*). The function transfers in our study adjusted only income, an argument, and assumed that the coefficients are constant over the 17 years.

In addition, it is notable that non-coastal residents' mean PTE and its range for Project 2 are the highest for both value and function transfers. This is reasonable because neither the value nor function transfer reflected the loss of the seagrass bed, a change in the supply side. This indicates that, while welfare was not as sensitive to time even in the long term, it was sensitive to changes in the supply of ecosystem services (i.e., loss of seagrass beds for Project 2). This calls a further study on contextually relevant research with particular focus on supply-side changes. However, since Japan has been relatively stable in the socioeconomic sense, our findings do not rule out the importance of other contextual changes that affect the supply and demand of ecosystem services, such as income, demographics, perceptions of nature, and the preference structure of individuals, through learning procedures or cultural transmissions (*Skourtos, Kontogianni & Harrison, 2010*).

## Limitations and future research

Our study is subject to two major limitations in terms of its implications for conservation projects: a biased sample and context-dependent results. First, because of the significant changes in internet accessibility from 1998 (13.4%) to 2015 (83.0%), the sample attributes may differ enough to influence the results (see Supplementary Information S2 for descriptive statistics). Second, the overall stability of WTP could be attributed to the specific context of Japan, where there is little drastic social-ecological change affecting the supply and demand of ecosystem services. Therefore, the stability level found in this study might not be applicable to other areas characterized by drastic social-ecological changes, such as developing countries.

To derive more general implications for conservation projects, further studies on the evolution of preferences and development of methodology for protest response analyses are encouraged. Further, given the asset of previous one-time studies on WTP estimates in various contexts and time periods across the world (e.g., Ecosystem Services Valuation Database (ESVD); *Van Der Ploeg & De Groot, 2010*), a similar temporal study should be conducted by exploiting the asset to elicit more general findings about the long-term evolution of preferences for nature. The accumulation of such studies would allow us to construct a more temporal-sensitive database for the valuation of ecosystem services and conduct a better temporal and spatial benefit transfer. A caveat, however, is the availability of raw data used for past studies. In particular, data for protest

responses may be limited in their availability. Priority should, thus, be given to cases in which drastic social-ecological changes and adaptive conservation projects are expected. Furthermore, while our study used CVM, conjoint analysis, another stated preference method to measure WTP (*Louviere, Hensher & Swait, 2000*) may be promising. A conjoint analysis uses profiles of a conservation project with various attributes (e.g., degree of conservation, development, and cost) and elicits respondents' preferences for trade-offs among these attributes. This could be particularly useful when a project faces serious trade-offs such as conservation vs. development. In addition, a conjoint analysis can capture marginal changes in WTP caused by changes in attributes such as quantity of ecosystem services supplied, and thus, could better capture factors affecting supply and demand for ecosystem services.

As our study showed, protest responses could provide non-negligible information for a conservation project in a real-world context. Protest responses are by no means residual information. In fact, several attempts have been made to use them (*García-Llorente, Martín-López & Montes, 2011*; *Cunha-e-Sá et al., 2012*), although such analyses remain underdeveloped (*Brouwer & Martín-Ortega, 2012*; *Tobarra-González, 2015*).

Although beyond the scope of this paper, the readers should note that being based on economic theory, our study has a narrow focus relative to the broader spectrum of individuals' preferences for nature and their approaches to it. What WTP can capture is limited in scope and economics is not the only approach to capture the preferences regarding nature. First, WTP captures assigned values, not held ones (*Brown, 1984*). While held values represent certain types of behavior (e.g., loyalty), end states (e.g., freedom), and quality (e.g., beauty) that individuals value, assigned values are what individuals assign to a given change over alternative outcomes based on their preferences (*Segerson, 2017*). Second, economic theory is silent regarding the motivations underlying individuals' preferences (*Flores, 2017*). However, exploring the motivations behind their preferences may be insightful for designing and managing conservation projects. One promising approach is *Schwartz's (2012)* human value theory (HVT), which explains the motivational basis for both attitudes and behavior. *Hicks et al. (2015)* applied HVT to explore the motivations behind fishers' preferences for marine ecosystem services. Finally, while WTP captures individual preferences, assuming consumer sovereignty with short-term focus, sustainable values assuming community sovereignty with long-term focus should also be considered (*Costanza, 2000*; *Norton, Costanza & Bishop, 1998*). Since conservation projects typically involve long-term horizons, the preference inconsistencies between short- and long-term horizons should be eliminated (*Norton, Costanza & Bishop, 1998*). For example, *Norton, Costanza & Bishop (1998)* propose a two-tiered decision structure to eliminate such inconsistencies.

## CONCLUSIONS

This study investigated the evolution of preferences for conservation projects (i.e., welfare measured in WTP and other project aspects assessed by protest responses). We compared preferences for three conservation projects in the SIS, Japan, at two distant time points, 1998 and 2015. Owing to the lack of a unanimous protocol for protest response

analysis and its use for WTP estimate, we proposed a simple comprehensive analytic framework that integrates protest response and WTP analyses.

Protest responses provide useful information to render a project adaptive to changes in the social-ecological system, the SIS. For instance, the payment method should be reconsidered. The welfare obtained from the projects was stable over a 17-year period, except for non-coastal residents in the case of Project 2. This possibly reflects the factors influencing changes in the demand and supply of the ecosystem services. Since the PTEs for both value and functional transfers were smaller than those in previous studies, they can be considered temporarily transferable. A function transfer performs less than a value transfer because of the contextual similarity over time and the insufficient specification of functions for the transfer.

Further temporal studies are highly encouraged, with focus on locations where significant social-ecological changes have occurred or are expected to occur, such as in developing countries. These studies can contribute to not only the primary study site but also the accumulation and sophistication of ecosystem services database such as ESVD. Doing so will enable a better benefit transfer when time and budget are unavailable to conduct a primary study.

### Funding
This research was supported by the Environment Research and Technology Development Fund (S-13) of the Ministry of the Environment, Japan. The funders had no role in study design, data collection and analysis, decision to publish, or preparation of the manuscript.

### Grant Disclosures
The following grant information was disclosed by the authors:
Environment Research and Technology Development Fund (S-13) of the Ministry of the Environment, Japan.

### Competing Interests
The authors declare that they have no competing interests.

### Author Contributions
- Takuro Uehara conceived and designed the experiments, analyzed the data, prepared figures and/or tables, authored or reviewed drafts of the paper, approved the final draft.
- Takahiro Tsuge conceived and designed the experiments, analyzed the data, contributed reagents/materials/analysis tools, prepared figures and/or tables.
- Takahiro Ota conceived and designed the experiments, performed the experiments, contributed reagents/materials/analysis tools.

### Data Availability
The raw data are provided in the Supplemental File.

## Supplemental Information

Supplemental information for this article can be found online at http://dx.doi.org/10.7717/peerj.5366#supplemental-information.

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
