# Peer review of "Long-term evolution of preferences for conservation projects in the Seto Inland Sea, Japan: a comprehensive analytic framework"

_PeerJ, doi:10.7717/peerj.5366_

## Round 0.1 · original submission · Minor Revisions

The paper is very good, but could be improved by adding a couple of paragraphs about the broader context and alternative approaches, as suggested by one of the reviewers.

Reviewer 1 ·

Basic reporting

This paper is interesting.

Experimental design

It is OK.

Validity of the findings

OK.

Additional comments

This paper is worth to be published in the Journal after minor revisions.

Lines 2,15,86 Seto Inland Sea - the Seto Inland Sea
After line 384. From Table 1, protest increases for Projects 1,2 and 3, but WTP increases for only
Project 2. You have to discuss the reason of such difference between Project 2 and
Projects 1 and 3.
Line 471. What is ESVD?

·

Basic reporting

This paper addresses an important and under-studied problem in environmental and resource valuation: whether and how valuations of possible outcomes can be inter-temporally transferred. Economists have increasingly used transfers of study results across geographic space, but much less attention has been paid to the possible transfer of results regarding preferences across time. As authors point out, there are many situations in which long-term planning requires speculation regarding the possible evolution of preferences across time, and the general tendency in resource economics to assume static preferences raises deep and complex issues regarding the understanding and measurement of future preferences.

One reason this topic is understudied is that, by its nature, empirical study of these questions requires series of data obtained at disparate times, times usually greater than the duration of most studies (which are often limited by funding sources and other factors). The authors provide a significant service by presenting (quasi) panel data that allow comparisons across a period of time. I applaud this work and the call for replications based on other cases with similar data (if available).

One minor question: At line 239, it is written: "Similarly, the rate of part-time workers, unemployed, and housewives who did not have good internet access was 36% in 2015 but 3% in 1998." This seems counter-intuitive to me. Is this correct? Given the increased penetration of internet access, one would expect the opposite.

Experimental design

Before I continue with this review, a caveat: I am not an economist, though I value what I have learned in working with economists. I will not presume to pass judgment on the technical and measurement aspects of this paper other than to note with plaudits the careful and thoughtful treatment of protest answers. I am assured that other readers will check this work carefully. My role is rather to look at the paper from the broader perspective of a philosopher and sustainability theorist; the remainder of this review should be understood in this light.

Validity of the findings

At lines 429-439, authors call for more valuation studies of intertemporal preferences for ecosystem services. This is an important recommendation. However, I would also argue that we need some much deeper thinking about how to understand and value resources and changes to natural and artificial systems across time. This paper is myopic in assuming that methods developed for one-time assessments of preferences as a measure of value can simply be applied by using (to the extent possible) these methods in temporal series. Those not so schooled in economic methodology, such as myself, might be quite skeptical of using a methodology that begins by assuming preferences are fixed, and then tries to measure how preferences change across time.

For all its strengths in using technical methods to study an important problem in resource and environmental valuation, the paper rests on a very shaky theoretical base. This may be inevitable given the current level of understanding and research by economists on this topic, but it ignores a significant literature on the topic of how to understand intertemporal relationships and perpetuates the isolation of economists from broader discussions of important problems having to do with the impacts of current activities on the future. That literature includes, for example:

B. G. Norton, “Intergenerational Equity and Environmental Policy: A Model Using Rawls’ Veil of Ignorance,” Ecological Economics, Vol. 1, pg. 137-159, 1989.

Brian Barry, "Sustainability and Future Generations," in Dobson, Fairness and Futurity, Oxford U. Press, 1999, 2003.

Brian Barry, "Inergenerational Justice" Stanford Encyclopedia of Philosophy. 2003, revised, 2015.

B.G. Norton, R. Costanza, and Bishop, R., “The Evolution of Preferences: Why ‘Sovereign’ Preferences May Not Lead to Sustainable Policies and What to Do about It,” Ecological Economics, 24(1998): pp. 193-212.

B.G. Norton, Sustainable Values, Sustainable Change: A Guide to Environmental Decision Making. U. of Chicago Press, 2015

It would, of course, be asking for a very different paper to expect that all of the issues raised in this broader literature in the present paper. I am not asking for this--assuming the technical aspects of the paper pass muster, I support publication without major revisions. But it would be a contribution to the broader intellectual process if the authors were to demonstrate awareness of these broader issues by adding appropriate references and by locating their economic approach to these problems in a broader context. For example, authors mention in passing that most economic studies (with much shorter duration) assume/profess that individual preferences are fixed. This is an apparent empirical assumption that is contradicted by the rapid change of, for example, U.S. attitudes toward smoking and LGBT marriage. In fact, of course, its a methodological assumption that is useful in making short-term economic comparisons. But to fail to even note that this paper exploits this assumption, and that there are alternatives in related literatures is to perpetuate the intellectual isolation of economists.

Additional comments

In summary, then, this paper makes a significant contribution in that it attacks a neglected topic and presents a case study that, in the context of its presentation, provides a clear example of how, given interesting data gathered in studies separated by 17 years, it is possible to use economic methodology to better understand some aspects of the evolution of preferences. We can all learn from this. But the paper, as it stands, does not speak to non-economists, nor does it encourage economists to see the paper as one interesting possibility among many for thinking about how we should evaluate the changes we cause, especially how we should weight changes that might seriously affect the future. While I do not advocate a major rewrite of the paper, I do think a couple of paragraphs setting the broader context and recognizing that there may be alternative approaches, would broaden the interest and also the reach of the paper.

Reviewer 3 ·

Basic reporting

The paper is clearly written and well understood; literature references and background information is provided to a satisfactory degree; the structure is professional following the state-of-the-art and raw data are shared; the paper id self-contained.

Experimental design

This research is within the aims and scope of the journal and well defined with a very good explanation of the corresponding knowledge gap; the investigation os rigorous; the methods have been described with enough detail and information to replicate.

Validity of the findings

No comment

Additional comments

no comment

---

## Round 0.2 · accepted · Accept

I can confirm that your revisions are sufficient and the paper is now accepted.

#